# Not just a Snapshot: An Italian Longitudinal Evaluation of Stability of Gut Microbiota Findings in Parkinson’s Disease

**DOI:** 10.3390/brainsci12060739

**Published:** 2022-06-04

**Authors:** Rocco Cerroni, Daniele Pietrucci, Adelaide Teofani, Giovanni Chillemi, Claudio Liguori, Mariangela Pierantozzi, Valeria Unida, Sidorela Selmani, Nicola Biagio Mercuri, Alessandro Stefani

**Affiliations:** 1UOSD Parkinson’s Center, Department of Systems Medicine, University of Rome Tor Vergata, 00133 Rome, Italy; dott.claudioliguori@yahoo.it (C.L.); pierantozzim@gmail.com (M.P.); stefani@uniroma2.it (A.S.); 2Department for Innovation in Biological, Agro-Food and Forest Systems (DIBAF), University of Tuscia, 01100 Viterbo, Italy; daniele.pietrucci@unitus.it (D.P.); gchillemi@unitus.it (G.C.); 3Institute of Biomembranes, Bioenergetics and Molecular Biotechnologies, IBIOM, Consiglio Nazionale della Ricerca (CNR), 70126 Bari, Italy; 4Department of Biology, University of Rome Tor Vergata, 00133 Rome, Italy; adelaide.teofani@uniroma2.it (A.T.); valeria.unida@gmail.com (V.U.); 5Department of Neuroscience, Unizkm, 1001 Tirana, Albania; sidorela.selmani@live.com; 6Department of Systems Medicine, University of Rome Tor Vergata, 00133 Rome, Italy; mercurin@med.uniroma2.it; 7Istituto di Ricovero e Cura a Carattere Scientifico (IRCCS) “Fondazione Santa Lucia”, 00179 Rome, Italy

**Keywords:** Parkinson’s disease, gut microbiota, gut–brain axis, dysbiosis

## Abstract

Most research analyzed gut-microbiota alterations in Parkinson’s disease (PD) through cross-sectional studies, as single snapshots, without considering the time factor to either confirm methods and findings or observe longitudinal variations. In this study, we introduce the time factor by comparing gut-microbiota composition in 18 PD patients and 13 healthy controls (HC) at baseline and at least 1 year later, also considering PD clinical features. PD patients and HC underwent a fecal sampling at baseline and at a follow-up appointment. Fecal samples underwent sequencing and 16S rRNA amplicons analysis. Patients’clinical features were valued through Hoehn&Yahr (H&Y) staging-scale and Movement Disorder Society Unified PD Rating Scale (MDS-UPDRS) Part-III. Results demonstrated stability in microbiota findings in both PD patients and HC over a period of 14 months: both alfa and beta diversity were maintained in PD patients and HC over the observation period. In addition, differences in microbiota composition between PD patients and HC remained stable over the time period. Moreover, during the same period, patients did not experience any worsening of either staging or motor impairment. Our findings, highlighting the stability and reproducibility of the method, correlate clinical and microbiota stability over time and open the scenario to more extensive longitudinal evaluations.

## 1. Introduction

The concept of “gut–brain axis”, a bidirectional channel of influence and communication between the brain and the enteric nervous system (ENS), was first introduced in the late 2000s [1]. This axis, sustained by neurons of the sympathetic and parasympathetic nervous systems, as well as by circulating hormones, neuro-modulatory molecules, and stress-related gastrointestinal mediators, has been linked to inflammatory bowel diseases, neuropsychiatric syndromes, and neurodegenerative diseases [2]. Later evidence rightfully included the gut microbiota in this axis, turning it into the “microbiota–gut–brain axis”; indeed, resident bacteria may influence the gut and, consequently, the brain [3]. On the other hand, metabolites produced by the brain can influence the gut and, consequently, the gut microbiota [4]. The gut microbiota is an heterogenous and symbiotic community composed of about 100 trillion bacteria inhabiting the human intestine, which influences both intestinal physiology and dysfunctional processes through its metabolic activities and host interactions [5]. The whole genetic asset of gut microbiota, called a microbiome, contains more than 3 million unique genes, 150 fold the number of human genes [6]. A dysbiotic gut microbiota may influence the progression of central nervous system (CNS) diseases [7]. The role of the gut microbiota has been observed in multiple sclerosis [8,9]**,** Alzheimer’s disease [10], autism spectrum disorder [11], and depression [12]. The involvement of the gut microbiota was observed in Parkinson’s disease (PD) using animal models [13] and through clinical observations; Braak and co-authors first postulated the presence of a “*noxa patogena*” ascending from the intestine or nasal cavity [14,15]. Today, dysbiosis of gut microbiota is commonly considered a well-established non-motor feature in PD, participating in both disease pathogenesis and, likely, clinical presentation [16].

In the last years, several studies investigated the specific role of altered gut microbiota in PD [17]. Different research groups, frequently utilizing fecal samples and sequencing, documented significant differences in gut microbial composition between PD patients and healthy subjects [18]. Scientific evidence from different parts of the world showed that PD patients have a higher relative abundance of bacteria from the genera *Akkermansia*, *Lactobacillus*, and *Bifidobacterium* and lower relative abundances of *Prevotella*, *Faecalibacterium*, *Bacteroidetes*, and *Blautia genera* [19,20,21,22]. Other groups found differences in microbiota composition between healthy subjects and PD patients, as well as between individuals with different PD phenotypes of symptomology [23,24]. Amongst the most consistent results, it is of note the reduction in “homeostatic” short chain fatty acids (SCFA)-producing families, like Lachnospiraceae and Prevotellaceae, as well as the increase in pro-inflammatory lipopolysaccharides (LPS)-producing families, like Enterobacteriaceae and Verrucomicrobiaceae [25,26,27]. In line with these studies, our group also investigated the presence of gut dysbiosis in PD patients [28]: our previous data showed that Enterobacteriaceae, Lactobacillaceae, and Enterococcaceae families were more abundant in PD fecal samples, whilst Lachnospiraceae families were decreased in PD patients compared to healthy subjects. Although most studies in literature linked intestinal dysbiosis to the pathogenesis of PD, a causal liaison between a dysbiotic gut microbiota and the development of PD is still far from being established [16]. More intriguingly, almost all PD microbiota studies were *“single snapshots”*, namely cross-sectional studies investigating intestinal microbial composition in PD patients and healthy subjects at a specific time, although considering possible confounders; indeed, only a few studies showed a longitudinal approach.

In this study, we introduce the time factor; we compare gut microbiota composition in a group of PD patients and healthy control subjects at baseline and at least 1 year later, also considering confounding factors and patients’ clinical features.

Our aim is to investigate the stability and, thus, the reproducibility of gut microbial findings in PD patients in a relatively short time period, to give greater dignity to the differences highlighted, without neglecting clinical aspects.

## 2. Materials and Methods

### 2.1. Population and Study Design

We recruited, during scheduled visits, outpatients with PD afferent at the Movement Disorders Center of the Neurological Clinic of the University of Rome “Tor Vergata” between January 2017 and January 2020. All patients met the Movement Disorders Criteria for PD [29]; PD diagnosis was done by at least two board-certified neurologists.

Patients were also required to meet the following criteria: (1) no cognitive impairment, as defined by a Montreal Cognitive Assessment (MoCA) score above 25; (2) no chronic gastrointestinal (GI) disease, including malabsorption; (3) no clinical history of major intestinal surgery, gastric lesions, or gastric resection; and (4) complete agreement with the study design. Exclusion criteria were: (1) other concomitant neurologic and/or psychiatric diseases; (2) systemic and/or neurologic inflammatory, infectious, or autoimmune diseases; (3) atypical parkinsonism syndromes and vascular parkinsonism; (4) acute GI inflammatory diseases or any other GI disease in the last 26 weeks; (5) use of domperidone, or any drug potentially affecting GI motility and integrity in the last 12 weeks; (6) use, in the last 12 weeks, of pre-probiotics or therapy based upon steroids, nonsteroidal anti-inflammatory drugs, or antibiotics; and (7) anamnesis suggestive of GI cancer pathology. We also recruited a healthy controls (HC) group, composed mainly of patients’ cohabiting relatives or partners, to minimize lifestyle-confounding factors.

The exclusion criteria for HC were: (1) presence of any neurological disease; (2) presence of acute or chronic GI diseases both during the study and in medical history; (3) medical history of major intestinal surgery, gastric lesions, or gastric resection; and (4) use in the last 12 weeks of any drug potentially affecting GI function and integrity (steroids, NSAIDs, drugs with pro-kinetic anti-kinetic function on the intestinal motility, anti-acid drugs, pre/probiotics, antibiotics).

Patients and HC underwent a fecal sampling at baseline and at a follow-up appointment at least 12 months later. PD patients were clinically evaluated, both at the baseline and at the follow-up visit, through Hoehn & Yahr staging scale (H&Y) and Movement Disorders Society Unified Parkinson’s Disease Rating Scale Part III (MDS-UPDRS Part III), performed during the best ON time, to estimate disease progression and degree of motor impairment, respectively. All study participants gave written informed consent after receiving an extensive disclosure of the study purposes and risks. Ethics Committee of Fondazione PTV Policlinico Tor Vergata approved the trial (RS 73/18).

### 2.2. Sequencing and 16S rRNA Amplicons Analysis

Fecal samples were collected and analyzed as described in our previous study [28]. Briefly, fecal samples were collected using the pre-analytical sample processing (PSP) stool collection tubes. These tubes ensure the storage and the conservation of the DNA at ambient temperature for at least three months. Samples were processed following the manufacturer’s instructions during the timeframe that allows the extraction of a relevant DNA yield. First, stool samples were lysed at 95 °C, and PCR inhibitors and cell debris were removed. Samples were then treated with Proteinase K at 70 °C. Next, a DNA purification step was performed using a spin column system. Finally, the DNA was extracted and quantified using a NanoDrop spectrophotometer ND1000 (Termofisher, Waltham, Massachusetts, USA). The paired-end sequencing of the V3-V4 region of the 16S gene was performed using an Illumina MiSeq 2 × 300 bp (Illumina Inc., San Diego, CA, USA). Raw sequencing data are available from the Sequencing Read Archive (SRA) database [30] with BioProject ID: PRJNA510730. Bioinformatic data analysis was performed using the following software: Fatsqc vr. 0.11.9, Cutadapt vr. 2.9, and QIIME 2.0 [31]. Fastqc and Cutadapt were used to assess the read quality and the presence of Illumina sequencing adapter, respectively. The QIIME 2.0 pipeline removed chimeras from sequencing data, merged reads, and grouped 16S reads in amplicon sequence variants (ASVs). In detail, the QIIME pipeline was used to call the DADA2 [32] software to cluster the read in ASVs, and the taxonomy of representative sequences was assessed using the Q2-feature-classifier [31] and the SILVA database vr. 138 [33].

### 2.3. Statistical Analysis

Statistical analyses were performed in R vr. 3.5.3 (https://www.R-project.org/ accessed on 25 April 2022), using the following packages: vegan, phyloseq, and DESeq2. Phyloseq was used to import the file generated by the QIIME 2.0 pipeline in R, and DESeq2 was used to normalize samples according to McMurdie et al. [34]. The vegan package was used to compute richness (number of observed species, Chao1, and ACE indices) and alpha diversity indices (Shannon and Simpson indices). Phyloseq was used to evaluate four beta diversity indices (Bray-Curtis and Camberra distances, weighted and unweighted Unifrac). To search for differences in microbiota structure, a PERMANOVA test with 9999 permutations was performed. Statistical differences of taxa abundances between patients and controls over time were evaluated using a repeated-measure ANOVA. In detail, we used the following test: Taxa ~ Status + Time + Status:Time + Error (Individual_ID).

This model allows searching for taxa, which can change in abundance depending on the status (PD or HC), depending on the time of the measurement (the first measure or the follow-up), or a combination of both effects (status and time). The same model was also applied to alpha diversity metrics. All the p-values were corrected for multiple testing, and only tests with a false discovery rate less than 0.05 were considered statistically significant. In the case of significant bacterial families after the ANOVA test, we performed post-hoc comparisons to take into account differences among groups. Furthermore, to make a comparison between clinical data - Levodopa Equivalent Daily Dose (LEDD), H&Y and MDS-UPDRS Part III - at baseline and at follow-up, we used a paired t-test, after assessing the normal distribution of data using the Shapiro–Wilk test. The t-test was used to highlight differences in clinical data between the patients at the baseline and at the follow-up. The first group used in the test was represented by the clinical values of patients at the baseline. The second group was represented by the clinical values of the same patients at the follow-up.

## 3. Results

### 3.1. Recruitment and Demographics

We recruited 31 participants: 18 PD patients and 13 HC. PD patients and HC underwent a fecal sampling at baseline and at a follow-up appointment an average of 14 (SD ± 1.8) months later. PD patients also underwent a clinical examination performed by H&Y and MDS-UPDRS Part III scales at baseline and at the follow-up visit. The main demographic and clinical variables are reported in Table 1.

PD patients and HC did not differ in terms of gender and age. Levodopa equivalent daily dose (LEDD) did not change significantly at the follow-up from baseline (paired *t*-test *p*-value = 0.349).

### 3.2. Alfa and Beta Diversity

According to statistical analysis, microbial composition over time showed interesting results both in alfa diversity and in beta diversity. In our study, alpha diversity did not show any statistical differences at baseline and at follow-up in both PD patients and HC according to two different metrics (Shannon and Simpson indices). We performed a repeated-measure ANOVA in order to evaluate the differences between the PD patients and HC between the baseline and the follow-up. The ANOVA results did not show differences considering PD status (PD or HC), time (baseline and follow-up), or the combined effect of time and PD status. These results indicate that the gut microbiota alpha diversity did not show any significant differences after the follow-up (Figure 1). Even not significant, we observed a reduction of both indices in both patients and controls at the follow-up visit. Considering the richness indices (number of observed species, Chao 1, and ACE indices), we identified a significant effect related to the time, indicating a reduction of richness in both PD patients and HC at the follow-up visit (Figure 1). This result suggests a decrease in the number of species in both patients and controls, independent of the pathology.

Our results also showed that the beta diversity was maintained in both PD patients and HC over time. More specifically, using the Bray–Curtis dissimilarity, no detectable differences were identified when analyzing the gut microbiota combining the effect of time (follow-up and baseline) and pathological status (PD or HC). In fact, the PERMANOVA test was not significant when both the effects were considered (*p*-value > 0.05 with 9999 permutations) (Figure 2A). Furthermore, observing the heatmap reported in Figure 2B, no clear pattern can be identified regarding the difference in time. Instead, some samples of the same patients cluster together (i.e., sample id 2, 3, 15, 18, 20, 22), independent of the analysis time. This scheme indicates that the microbiota has remained almost stable during the baseline and the follow-up analyses. Moreover, the baseline differences between patients and HC were unchanged at follow-up.

### 3.3. Families

After observing the alpha and beta diversity results, we decided to perform the most detailed analysis to identify specific taxa whose abundance may vary between baseline and follow-up tests. The beta diversity indicated that the structure of the microbiota did not change over time; however, the alpha diversity showed a reduction of richness in both patients and controls at the follow-up visit. Consequently, we performed the repeated ANOVA test on the bacterial families to identify the taxa which could be responsible for this trend. Through the analysis of differential abundance, no statistically significant changes in the composition of microbial families emerged due to the combined effect of pathology and time. Only without considering the effect of the disease in the repeated-measure ANOVA test did we identify families changing over time in the same way in PD patients and HC. In total, six families decreased at the follow-up visit compared to the baseline in both patients and HC. These families belong mainly to the Phylum Bacteroidota (A. Bacteroidaceae, B. Rikenellaceae, C. Barnesiellaceae, D. Marinifilaceae, E. Tannerellaceae) but also to Firmicutes (F. Lachnospiraceae). The relative abundance of these families is reported in Figure 3 and Table 2. The relative abundance of two families also varied among groups, independent of time, suggesting a difference between groups. In detail, the Lachnospiraceae and Bacteroidaceae families were less abundant in PD patients compared to controls. Although these results were not significant after adjusting for multiple testing, they indicated a specific trend.

## 4. Discussion

Our study demonstrated stability in microbiota findings both in PD patients and in HC over a period of approximately 14 months. Not only did microbiota composition remain stable in both PD patients and HC but also, and more intriguingly, differences in microbiota composition between PD patients and HC remained stable over time.

Our results cover both alpha and beta diversity, as well as microbial families.

Alpha diversity is commonly considered an expression of the number of species in each sample; its stability in our data highlights how our microbiota analysis is effective in showing biodiversity over time. We did not observe differences in two different alpha diversity metrics at the baseline and follow-up. However, we have observed, in a recent meta-analysis, that the alpha diversity alone does not represent a good microbiota marker for PD [35]. Furthermore, de la Cuesta-Zuluaga J and colleagues showed that the alpha diversity reaches a plateau in elderly individuals [36]. Although not statistically significant, we observed a reduction for both Shannon and Simpson indices in both patients and controls. We performed the analysis by using the richness indices (number of observed species, Chao1, and ACE) and identified a reduction of richness in both patients and controls.

Conversely, beta diversity is an expression of gut microbiota community structure. In our study, this community structure was maintained in both PD patients and HC. Furthermore, the baseline differences between PD patients and HC remained unchanged at follow-up; hence we can assume that the composition in community structure of microbiota between PD and HC is preserved over time.

Regarding microbiota families, the analysis of the differential abundance did not find statistically significant changes in the composition of the families due to the combined effect of pathology and time; this data further substantiates the stability of the microbiota over time, which is greatly influenced, among other factors, by fetal and perinatal life [37]. By excluding the effect of the disease, we identified families that vary over time in the same way in patients and controls. This trend is probably attributable to the influence of external factors (lifestyle and diet changes) and does not affect the microbiota stability. For example, a reduction of the microbiota diversity was observed due to seasonality [38].

Finally, the trend observed in Lachnospiraceae and Bacteroidaceae families, to be less abundant in PD patients than in controls, highlights, once again, the presence of differences in the microbiota structure between PD patients and HC, as previously reported [28,35].

Our study also considered PD clinical aspects in terms of progression and motor deficits. Interestingly, during the observation period, patients did not experience any worsening either in terms of staging or in terms of motor impairment. Indeed, our data showed no significant differences in H&Y and MDS-UPDRS Part III scores between the baseline and follow-up visit, although this does not exclude minimal changes in single items of MDS-UPDRS-III. Furthermore, it should be noted that the relatively short observation period did not allow for remarkable changes to the therapeutic regimen. In our opinion, these results allow us to correlate clinical and microbiota stability over time, reinforcing the pathogenic role of gut microbiota change in PD patients. Otherwise, the presence of clinical deterioration during the observation period with an unaffected microbiota would have weakened this link. Our results are in line with a previous study by Aho and colleagues, in which the stability of the fecal microbiota was established after two years [39]. Of note is also the stability of microbiota findings in the HC group. Dealing with a microbiota that shows stability and reproducibility is the basis for each disease-modifying therapy targeting microbiota.

The main objective of our research was to demonstrate the stability of the microbiota composition over a short time and the reproducibility of the method of analysis; therefore, data from other longitudinal studies investigating disease progression related to microbiota changes are not comparable [40]. However, the stability of both gut microbiota composition and clinical features in PD patients over a relatively short time period does not exclude that microbiota may vary over a longer period of time or with the worsening of the disease and/or with the intervention of a trigger factor.

The main limitations of this study are the relatively short observation period and the sample size, which was sufficient to make inferences but which would have had a better impact if larger. However, although small, our cohort is characterized by rather homogeneous characteristics. Looking ahead, our study lays the foundation for longitudinal evaluations with a wider observation time window and a larger cohort of patients, to correlate any worsening of the disease with further alterations of the microbiota over time.

## 5. Conclusions

In our study, we do not identify differences in the gut microbiota (beta diversity) structure at the follow-up in both PD patients and HC, which remained stable for both patients and controls. These results suggest that the gut microbiota may remain stable over a period of 14 months. We identified a reduction in some richness indices, in both patients and controls, indicating that some species may reduce their abundance in the gut microbiota. We confirmed this analysis with a differential abundance test on the bacterial families using a repeated-measure ANOVA. Seven families showed a decreasing trend in PD patients, which was also identified in HC. Consequently, these differences may be caused by other external factors (i.e., alimentations) unrelated to PD progression. Moreover, the PD cohort showed, at follow-up, the same degree of disease progression (H&Y) and motor impairment (MDS-UPDRS Part III) with respect to the baseline, supporting the idea that, in PD, microbial stability and disease stability could be correlated. Dysbiosis of gut microbiota can be considered a relevant and reliable feature of PD that can provide insights into the disease pathophysiology. For future studies, a crucial step will be the fecal sampling and analysis at multiple timepoints in disease progression, a wider observation window, and a larger patient population with different H&Y and MDS-UPDRS Part III scores. These longitudinal data could identify physio-pathological correlations between the variation in microbiota composition and the PD progression. Last but not least, these recent results on gut microbiota stability create an opportunity for new studies aimed at understanding whether and to what extent therapeutic interventions (levodopa, iCOMT, and/or the initiation of advanced therapies such as Levodopa Carbidopa Intestinal gel) can play an active role in modifying the gut microbiota.

## Figures and Tables

**Figure 1 brainsci-12-00739-f001:**
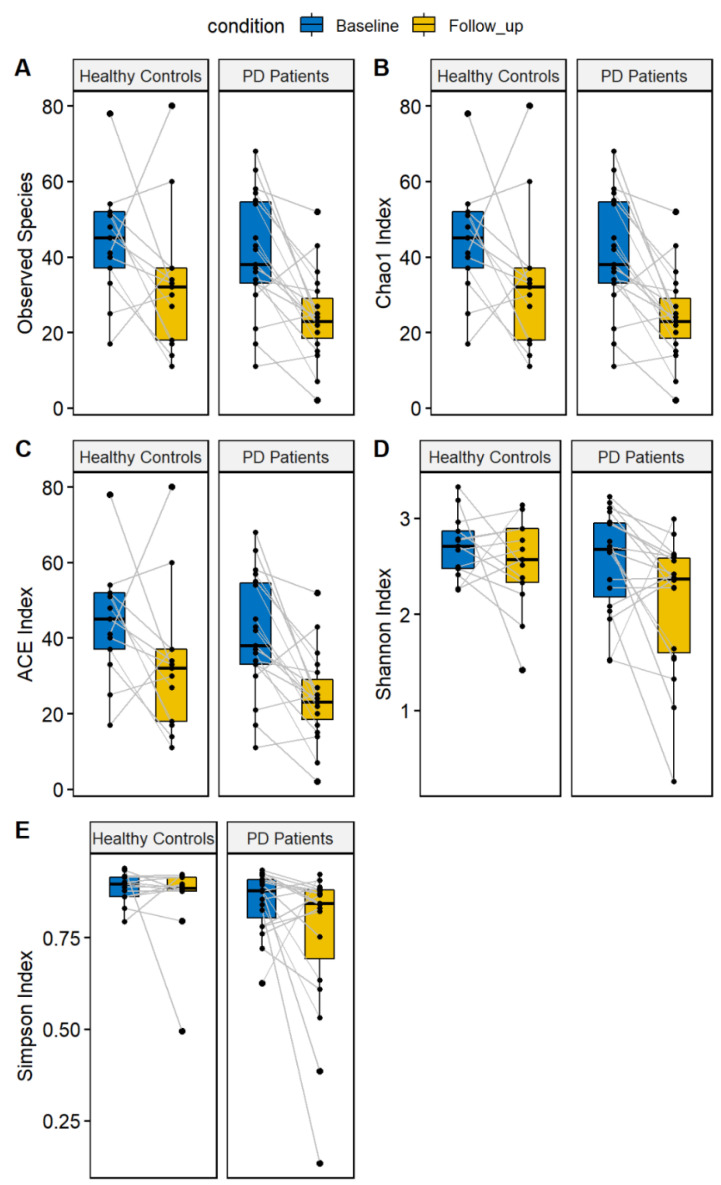
Alpha diversity indices evaluated for Parkinson’s disease patients and healthy controls. The following indices are reported: number of observed species (**A**), Chao1 index (**B**), ACE index (**C**), Shannon Index (**D**), and Simpson Index (**E**). For both patients and controls, the values at the first visit (“baseline”, colored in blue) and the follow-up (colored in yellow) are reported.

**Figure 2 brainsci-12-00739-f002:**
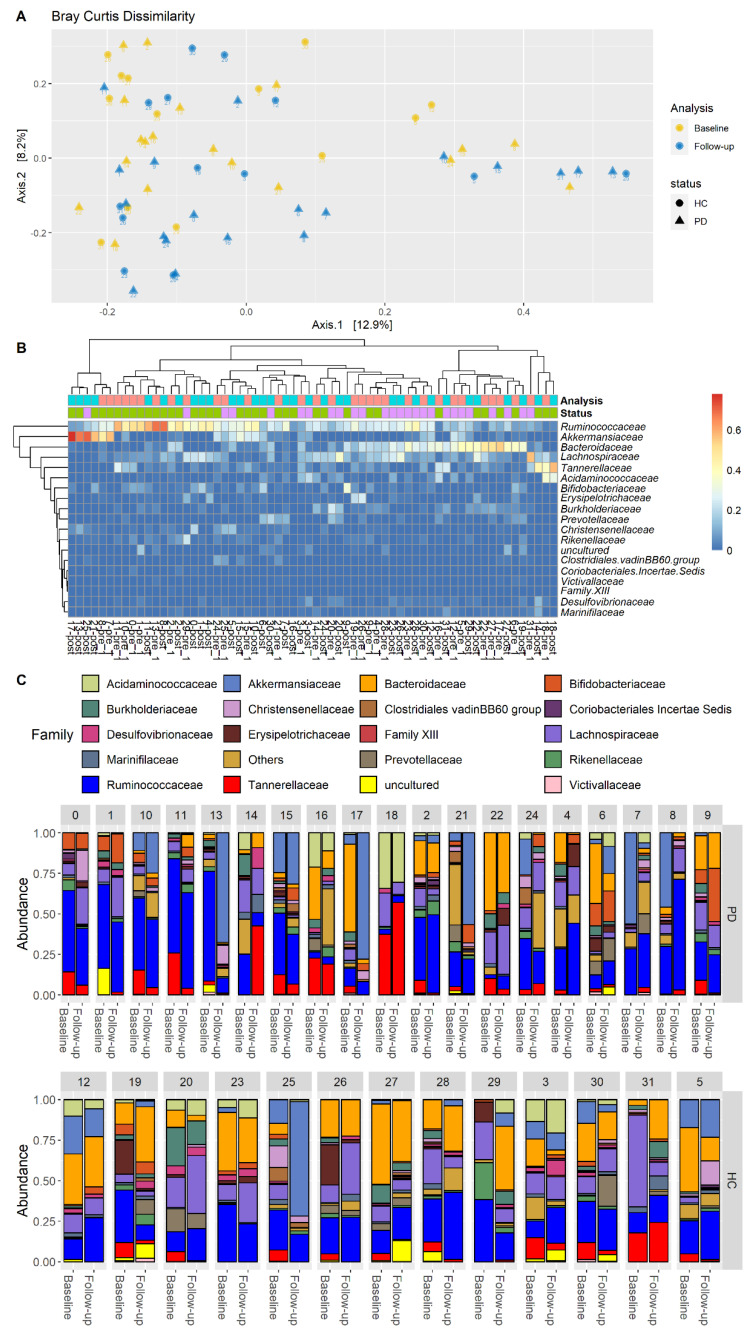
Beta diversity evaluated for Parkinson’s disease patients and healthy controls. (**A**) Principal coordinate analysis. Each patient is represented by an anonymized ID; the shape indicates the microbiota evaluation (circle = baseline, triangle = follow-up). (**B**) Heatmap of Bray–Curtis distances between patients (columns) and the main bacterial families (rows). The cell reports the relative abundance of each family in all samples. Samples are reported in columns using the following color scheme: PD = green, HC = purple, baseline = cyan, follow-up = pink. (**C**) Barplot representing the relative abundance of microbial families across all samples. Each subject is represented with two barplots, one for the baseline and one for the follow-up test.

**Figure 3 brainsci-12-00739-f003:**
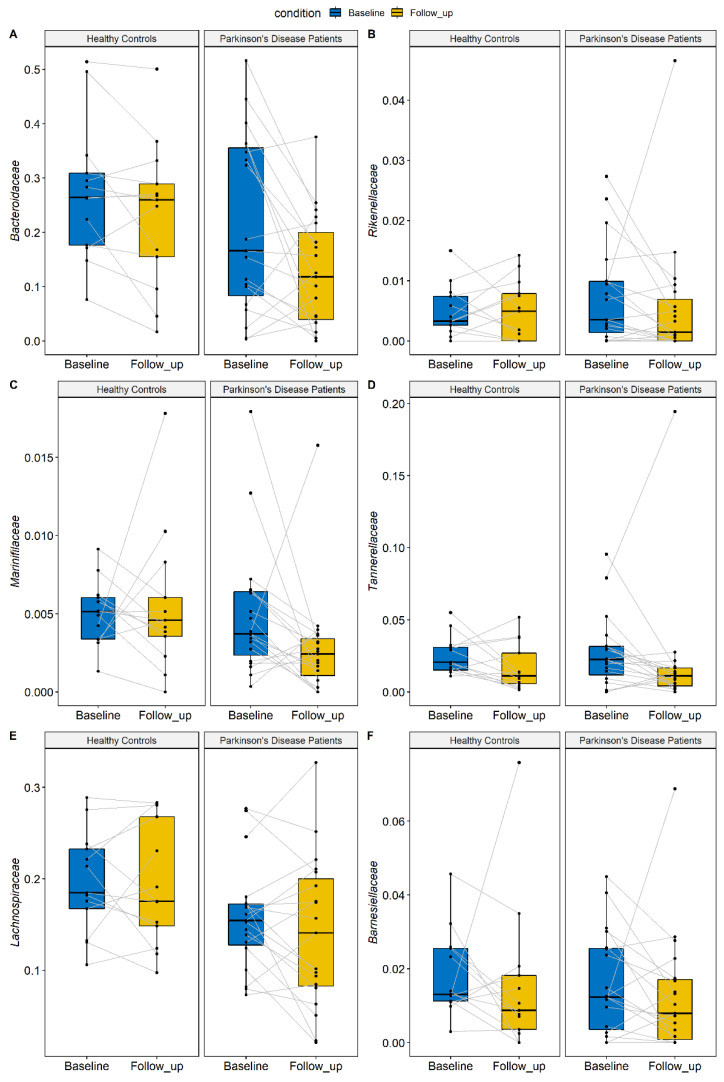
Taxa identified by the repeated-measure ANOVA test. All the families are reduced in both patients and controls at the follow-up (colored in yellow) compared to the baseline (colored in blue), independent of the health status (PD or HC). The effect of time is statistically significant after performing a multiple test correction.

**Table 1 brainsci-12-00739-t001:** Demographic and clinical data of our population.

		PD	HC
**Tot**		18	13
**Sex (F/M) ** ^ **1** ^		08/10	06/07
**Age (years)**		63.5 ± 8.1	62.8 ± 7.8
**Dis. Duration (months)**	**Baseline**	35.7 ± 10	
**H&Y**	**Baseline**	2.05 ± 0.6	
**H&Y**	**Follow-up**	2.15 ± 0.5	
**MDS UPDRS-III**	**Baseline**	23.6 ± 6.2	
**MDS UPDRS-III**	**Follow-up**	24.3 ± 5.6	
**LEDD**	**Baseline**	450.3 ± 40.2	
**LEDD**	**Follow-up**	479.2 ± 25.4	

^1^ Abbreviations: M = male; F = female; H&Y = Hoehn & Yahr staging score; MDS UPDRS-III = Movement Disorders Society Unified Parkinson’s Disease Rating Sale Part III score; LEDD = Levodopa Equivalent Daily Dose.

**Table 2 brainsci-12-00739-t002:** Bacterial families identified by the repeated-measure ANOVA test. The relative abundance is reported for HC and PD patients at baseline and follow-up tests. The *p*-values reported refer to the significance between HC and PD patients (status), follow-up and baseline (time), or considering the effect of status and time. All the *p*-values are corrected for multiple testing.

	Abundance at Baseline		Abundance at Follow-up		Repeated-Measure ANOVA Results		
	HC	PD	HC	PD	*p*-Value Status	*p*-Value Time	*p*-Value Status: Time
*Bacteroidaceae*	2.74 × 10^−1^	2.15 × 10^−1^	2.32 × 10^−1^	1.28 × 10^−1^	8.16 × 10^−2^	5.99 × 10^−3^ *	9.80 × 10^−1^
*Tannerellaceae*	2.58 × 10^−2^	2.74 × 10^−2^	1.71 × 10^−2^	2.21 × 10^−2^	1.00 × 10^0^	2.12 × 10^−2^ *	9.80 × 10^−1^
*Rikenellaceae*	4.95 × 10^−3^	7.55 × 10^−3^	5.02 × 10^−3^	5.85 × 10^−3^	8.16 × 10^−2^	2.12 × 10^−2^ *	9.82 × 10^−1^
*Marinifilaceae*	5.04 × 10^−3^	4.98 × 10^−3^	5.51 × 10^−3^	2.81 × 10^−3^	8.22 × 10^−1^	2.65 × 10^−2^ *	9.89 × 10^−1^
*Lachnospiraceae*	1.96 × 10^−1^	1.58 × 10^−1^	1.94 × 10^−1^	1.41 × 10^−1^	8.45 × 10^−2^	1.23 × 10^−3^ *	9.80 × 10^−1^
*Barnesiellaceae*	1.85 × 10^−2^	1.60 × 10^−2^	1.57 × 10^−2^	1.29 × 10^−2^	6.60 × 10^−1^	1.11 × 10^−2^ *	9.81 × 10^−1^

Finally, we processed data regarding the clinical aspects of PD patients over time. The analysis of patients’ clinical variables showed no significant differences in H&Y (paired *t*-test *p*-value = 0.349) and MDS-UPDRS Part III (paired *t*-test *p*-value = 0.395) scales between baseline and follow-up visit scores.“*” indicates a *p*-value lower than 0.05.

## Data Availability

Raw sequencing data are available at Sequencing Read Archive (SRA) database with BioProject ID: PRJNA510730.

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
