# Peer review of "Not just a Snapshot: An Italian Longitudinal Evaluation of Stability of Gut Microbiota Findings in Parkinson’s Disease"

_brainsci, 2022, doi:10.3390/brainsci12060739_

Round 1

Reviewer 1 Report

  1. The title should mention the article type (study design and localization). It is advised to avoid ‘‘:’’ due to the searching terms' purpose.
  2. Abstract. It is lacking the number of individuals in the control and cases.
  3. There are some grammatical errors throughout the manuscript. E.g. long phrases, misspellings.
  4. It is advised to remove all grouped references. E.g. ‘‘patients and healthy subjects [3-16];’’
  5. Introduction is missing important aspects about the ‘‘gut-brain axis’’. Some highlights that could be included: Who first mentioned? Who first associated with PD? How was this axis assessed in other groups of neurological conditions? Could the authors provide some percentages of gut microbiota?
  6. PD diagnosis. Was the diagnosis of PD done by at least two board-certified neurologists?
  7. What were the drugs included in the ‘‘any drug potentially affecting gastrointestinal function’’ list?
  8. Scales. Who performed the H&Y and MDS-UPDRS Part III? Was a previous request for their use done?
  9. Statistics. Could the authors describe the data distribution? Was done any analysis using the scales (or were they used only to show disease progression)? Could the authors describe the model in more detail? Was T-test used to compare an individual or a group of individuals?
  10. Table 2. Does the description of table 2 only include the significant data? If no, an ‘‘*’’ should be written in those with significant data.
  11. What is the authors’ opinion about the low number of participants in the study? Do they believe that the data encountered could not be affected by the ‘‘n’’?
  12. What is the significance of providing evidence that the microbiota will not change over time? Could the microbiota only be related to development as a trigger factor? The authors should highlight this in the discussion.

Author Response

Response to Reviewer 1 Comments

We would like to thank the Reviewer 1 for the careful and inspiring work

Point 1:          The title should mention the article type (study design and localization). It is advised to avoid ‘‘:’’ due to the searching terms' purpose.

Response 1: We thank the Reviewer for the suggestion. So, we changed the title from “Gut Microbiota alterations in Parkinson’s Disease: not just a snapshot” to “Not just a snapshot. An Italian longitudinal evaluation of stability of gut Microbiota findings in Parkinson’s Disease”  

Point 2: Abstract. It is lacking the number of individuals in the control and cases.

Response 2: We thank the Reviewer for the detection. We introduced in the abstract the number of PD patients and controls.

Point 3: There are some grammatical errors throughout the manuscript. E.g. long phrases, misspellings.

Response 3: We apologize for grammatical errors. We tried to correct misspellings and to reduce long phrases.  

Point 4: It is advised to remove all grouped references. E.g. ‘‘patients and healthy subjects [3-16];’’

Response 4: We thank the Reviewer for the opportunity to clarify this point. MDPI journals usually allow multiple references. In this case, our aim was to provide insights and to frame the most evidence about microbiota changes in PD, without going into too much detail. Indeed, a more articulated treatment would not be in line with the aim of the study oriented to investigate the stability of the microbiota over the time and not to summarize all the previous results as a review

Nevertheless, we tried to reduce them and distribute them.

Point 5: Introduction is missing important aspects about the ‘‘gut-brain axis’’. Some highlights that could be included: Who first mentioned? Who first associated with PD? How was this axis assessed in other groups of neurological conditions? Could the authors provide some percentages of gut microbiota?

Response 5: We thank the Reviewer for the suggestion. So, we modified the introduction by inserting both the concept of gut brain axis and more details about microbiota. 

Point 6: PD diagnosis. Was the diagnosis of PD done by at least two board-certified neurologists?

Response 6: We thank the Reviewer for the opportunity to clarify this point. Our Parkinson’s Centre includes several neurologists with different tasks. We are a heterogeneous group, in which some are dedicated to clinics, others to research, others to clinical trials. Therefore, each patient is always evaluated by at least 2 neurologists. According to these considerations we modified the text by inserting the phrase “PD diagnosis was done by at least two board-certified neurologists”  

Point 7: What were the drugs included in the ‘‘any drug potentially affecting gastrointestinal function’’ list?

Response 7: We excluded steroids, NSAIDs, drugs with pro-kinetic anti-kinetic function on the intestinal motility, anti-acid drugs, pre/probiotics, antibiotics. We inserted this information in the text.   

Point 8:          Scales. Who performed the H&Y and MDS-UPDRS Part III? Was a previous request for their use done?

Response 8: H&Y and MDS UPDRS Part III are routinely performed to all PD outpatients as a common clinical practice; we only scheduled the visits taking into account the time needed for follow-up. 

Point 9:          Statistics. Could the authors describe the data distribution? Was done any analysis using the scales (or were they used only to show disease progression)? Could the authors describe the model in more detail? Was T-test used to compare an individual or a group of individuals?

Response 9: We checked the distribution of the data using a Shapiro-Wilk test in R, that assessed a normal distribution of the clincal data. The t-test was used to compare two groups of individuals. The first group was represented by the patients at the baseline and the second one the patients at the follow-up. The t-test was paired, in order to take into account the individual variability. We added these details in the materials and methods section.

Point 10:        Table 2. Does the description of table 2 only include the significant data? If no, an ‘‘*’’ should be written in those with significant data.

Response 10: Yes, after performing a multiple testing correction, all the bacteria reported in Table 2 are significant data, considering the “Time” variable. Since all the taxa can be evaluated in the Repeated Measure Anova using three variables (“Time”, “Status” or the combined effect between Time and Status, indicated as “Time:Status”), it’s difficoult to add a “*” near the name of the taxa. However, in the table are reported the p-values, so we decided to report “*” near the p-value. We added also the information in the Table footnote and in Figure 3

Point 11:        What is the authors’ opinion about the low number of participants in the study? Do they believe that the data encountered could not be affected by the ‘‘n’’?

Response 11: We thank the Reviewer for the opportunity to clarify this point. As we wrote in the paper, we think that the relatively small number of participants open the scenario to further evaluation with a wider simple size, to confirm the interesting data highlighted in our work. However, albeit small, our cohort are characterized by rather homogeneous features (we added this concept in the discussion).

Point 12:        What is the significance of providing evidence that the microbiota will not change over time? Could the microbiota only be related to development as a trigger factor? The authors should highlight this in the di-scussion.

Response 12: We thank the Reviewer for the observation. The aim of our study is to demonstrate stability of microbiota in a relatively short time period, to emphasize the reproducibility of the results in the same population rather than being the result of chance, giving more strength to our research. The possibility to deal with a relatively stable microbiota community also represents the basis for disease-modifying therapies targeting microbiota. It’s also interesting the suggestion of a trigger factor, so we inserted it in the discussion 

Reviewer 2 Report

Since only Shannon and Simpson indices were used in Alpha Diversity, Observed, Chao1, and ACE indices are required.

Since it is said that there is no difference in the composition of the microbial community only with the α-diversity result, a bar graph showing the composition of the microbial community should be provided.

A comparison of alpha and beta diversity between follow-up of healthy controls and follow-up of PD patients should be provided because healthy controls do not appear to change over time, but PD patients tend to have decreased alpha diversity over time. 

Author Response

Response to Reviewer 2 Comments

We would like to thank the Reviewer 2 for the careful and inspiring work

Point 1: Since only Shannon and Simpson indices were used in Alpha Diversity, Observed, Chao1, and ACE indices are required.

Response 1: As the reviewer suggested we performed the suggested analysis. The Material and Methods section, the Figure 1 and the Discussion were modified accordingly.

Point 2: Since it is said that there is no difference in the composition of the microbial community only with the α-diversity result, a bar graph showing the composition of the microbial community should be provided.

Response 2: As the reviewer suggested we added a bargraph in Figure 3

Point 3: A comparison of alpha and beta diversity between follow-up of healthy controls and follow-up of PD patients should be provided because healthy controls do not appear to change over time, but PD patients tend to have decreased alpha diversity over time. 

Response 3: The reviewer is right, we described better the ANOVA test results in the chapter “3.2 Alfa and Beta-diversity”, also adding the indices suggested in the first comment. We found a reduction of richness indices (Chao1, ACE and number of observed species) in both patients and controls at follow up visit. The same trend can be observed for Simpson and Shannon indices, although not statistically significant. However, we didn’t find any statistical difference between PD and HC, or any statistical difference between PD and HC combined with the Time (effect baseline or follow-up). This is in line with the results of the six families reported in Figure 3. This result indicates that there are some differences between the two time points, however, since they happen in both groups. Consequently, these microbiota changes can’t be linked to the pathology, but to some external factors (such as nutrition). These considerations were better explained in the results and discussion section.

Regarding the beta-diversity, we performed a PERMANOVA analysis to the dataset. The results were not significant with 9999 permutations. Consequently, the structure of the microbiota didn’t change for PD and HC after 12-14 months.
